# Glycyrrhizic Acid Prevents Paclitaxel-Induced Neuropathy via Inhibition of OATP-Mediated Neuronal Uptake

**DOI:** 10.3390/cells12091249

**Published:** 2023-04-25

**Authors:** Ines Klein, Jörg Isensee, Martin H. J. Wiesen, Thomas Imhof, Meike K. Wassermann, Carsten Müller, Tim Hucho, Manuel Koch, Helmar C. Lehmann

**Affiliations:** 1Department of Neurology, Faculty of Medicine and University Hospital Cologne, University of Cologne, 50931 Cologne, Germanyhelmar.lehmann@uk-koeln.de (H.C.L.); 2Center for Molecular Medicine Cologne (CMMC), Faculty of Medicine and University Hospital Cologne, University of Cologne, 50931 Cologne, Germany; 3Translational Pain Research, Department of Anaesthesiology and Intensive Care Medicine, Faculty of Medicine and University Hospital Cologne, University of Cologne, 50931 Cologne, Germany; 4Pharmacology at the Laboratory Diagnostics Center, Therapeutic Drug Monitoring, Faculty of Medicine and University Hospital Cologne, University of Cologne, 50931 Cologne, Germany; 5Center for Biochemistry, Institute for Dental Research and Oral Musculoskeletal Research, Faculty of Medicine and University Hospital Cologne, University of Cologne, 50931 Cologne, Germany; 6Department of Neurology, Hospital Leverkusen, 51375 Leverkusen, Germany

**Keywords:** paclitaxel, taxol, CIPN, OATP, OATP1B2, OATP1A1, glycyrrhizic acid, drug transport

## Abstract

Peripheral neuropathy is a common side effect of cancer treatment with paclitaxel. The mechanisms by which paclitaxel is transported into neurons, which are essential for preventing neuropathy, are not well understood. We studied the uptake mechanisms of paclitaxel into neurons using inhibitors for endocytosis, autophagy, organic anion-transporting polypeptide (OATP) drug transporters, and derivatives of paclitaxel. RT-qPCR was used to investigate the expression levels of OATPs in different neuronal tissues and cell lines. OATP transporters were pharmacologically inhibited or modulated by overexpression and CRISPR/Cas9-knock-out to investigate paclitaxel transport in neurons. Through these experiments, we identified OATP1A1 and OATP1B2 as the primary neuronal transporters for paclitaxel. In vitro inhibition of OATP1A1 and OAT1B2 by glycyrrhizic acid attenuated neurotoxicity, while paclitaxel’s antineoplastic effects were sustained in cancer cell lines. In vivo, glycyrrhizic acid prevented paclitaxel-induced toxicity and improved behavioral and electrophysiological measures. This study indicates that a set of OATPs are involved in paclitaxel transport into neurons. The inhibition of OATP1A1 and OATP1B2 holds a promising strategy to prevent paclitaxel-induced peripheral neuropathy.

## 1. Introduction

Peripheral neuropathy is one of the most common side effects of the microtubule-stabilizing chemotherapeutic agent paclitaxel, affecting up to 87% of all treated patients [1,2]. Currently, paclitaxel and its derivatives, such as docetaxel and cephalomannine, are most commonly used to treat ovarian, breast, lung, prostate, head and neck, and gastric cancer entities [3]. In cancer cells, paclitaxel induces cell death via microtubule stabilization [4,5]; however, thanks to its microtubule-stabilizing actions, it is able to induce neuronal damage. Patients suffer predominantly from sensory symptoms, such as pain and numbness in hands and feet because of paclitaxel accumulation in the dorsal root ganglia (DRG) [6]. As the DRGs possess a more permeable blood–nerve barrier, sensory neurons in the DRG are highly susceptible to paclitaxel accumulation [7]. The paclitaxel-induced length-dependent axonal sensory neuropathy correlates with the dose, infusion time, underlying conditions, and cotreatments with other drugs [8,9]. The influence of paclitaxel on A-β fibers gives rise to the main chemotherapy-induced peripheral neuropathy (CIPN) characteristics, such as numbness and loss of vibration sense. Dysesthesia and cold and mechanical hypersensitivity are due to the influence of paclitaxel on Aδ- and C-fibers [10].

Currently, the pathological cascade of how paclitaxel damages nerve fibers is incompletely understood. It is assumed that the multimodal effects of paclitaxel cause neuronal damage, such as axonal transport impairment, mitotoxicity, and inflammation [11,12]. In the peripheral nervous system (PNS), the microtubule stabilization affects protein, organelle, nutrient, neurotransmitter, and mRNA transport [13]. A following ATP undersupply in the periphery results in mitochondrial changes, such as swelling and vacuolization, and increased pain sensation [14]. Several inflammatory markers, such as interleukins and C-X-C motif chemokines, are thought to elicit pain sensations in patients [15]. Moreover, by attracting macrophages, neuronal degeneration is fostered [16].

Because of this plethora of toxic effects, there are currently no clinical treatments to prevent or treat paclitaxel-induced neuropathy. Furthermore, there is experimental evidence that different paclitaxel delivery mechanisms impact the kinetics of paclitaxel in nervous tissue and hence the degree of neurotoxicity [17,18]. With this study, we aim to intervene in the paclitaxel toxicity as early as possible, in this case on paclitaxel uptake into the neuronal cell. We therefore reasoned that the further characterization of the neuron-specific uptake of paclitaxel may provide novel strategies to prevent peripheral neuropathy.

It is assumed that paclitaxel is actively transported into cells by organic anion-transporting polypeptides (OATPs) [19,20,21]. These encompass a family of plasma membrane transporters essential for the absorption, clearance, and tissue distribution of endogenous compounds and xenobiotics. Corresponding genes are named with the prefix SLCO, while encoding proteins are marked with OATP. Whereas the expression pattern of OATP in cancer cells and hepatocytes is well known, in sensory neurons, the overall expression profile of OATP is less well studied. An exception to this is OATP1B2 (in rodents, corresponding to OATP1B1 and OATP1B3 in humans), whose pharmacological or genetic modulation ameliorated paclitaxel-induced neuropathy in a preclinical model [22]. However, symptoms were only partially alleviated, which indicates that other transporters are likely involved. Moreover, OATP1B3, the human homolog of OAT1B2, has been found in ovarian and associated cancer cell lines, such as ovarian cancer cells [23], colorectal and pancreatic cancer cells [24], and castration-resistant prostate cancer cells [25]. All of these are cancer entities that are treated with paclitaxel.

This leads us to our hypothesis that organic anion-transporting polypeptides (OATPs), which are essential for the absorption and tissue distribution of endogenous and exogenous compounds, are involved in paclitaxel uptake into sensory neurons. This study identified OATP1A1 and OATP1B2 as paclitaxel transporters in sensory neurons. Further in vivo experiments showed that inhibiting paclitaxel uptake with glycyrrhizic acid could alleviate paclitaxel-induced neuropathy. This study sheds light on the mechanism of paclitaxel-induced neuropathy and identifies potential targets for prevention and treatment.

## 2. Materials and Methods

### 2.1. In Vitro Experiments

#### 2.1.1. Cell Lines

All cells for cell culture experiments were maintained in a sterile incubator humidified with 95% air and 5% CO_2_ at 37 °C. The respective cell culture medium and concentration of fetal bovine serum (FBS) are listed in Table 1. All cell culture media were supplemented with 1% of penicillin-streptomycin. When cells reached 80–90% confluency, cells were divided into new flasks or well plates for experiments.

#### 2.1.2. Paclitaxel Accumulation

Intracellular paclitaxel accumulation in primary sensory neurons and F11 (ATCC Cat# PTA-11448, RRID:CVCL_0H91) cells was measured via (i) fluorescent paclitaxel exposure (P22310, Thermo Fisher Scientific, Waltham, MA, USA) or (ii) liquid chromatography coupled with tandem mass spectrometry (LC-MS/MS). Cells were either treated with increasing fluorescent paclitaxel concentrations or increasing concentrations of solvent-based Cremophor-ethanol (CreEL) paclitaxel. Cells were treated for 30 min, 1 h, or 2 h with the respective solutions. After incubation, cells were rinsed with PBS, and paclitaxel accumulation was determined by the plate reader at 540 nm when fluorescent paclitaxel was used or by LC-MS/MS when CreEL paclitaxel was used.

#### 2.1.3. Assessment of Different Uptake Mechanisms

To investigate the possible uptake mechanisms of paclitaxel, chemicals were used that inhibit clathrin-mediated endocytosis, caveolae-mediated endocytosis, macropinocytosis, autophagy, and phagocytosis (Table 2).

Further, 2.5 × 104 F11 cells were seeded in black 96-well plates and cultured in phenol red-free medium for 48 h. Chlorpromazine at 0.0064 µM, 0.016 µM, 4 µM, and 100 µM was applied for 1 h. Indomethacin at 75 µM, 150 µM, 225 µM, and 300 µM was applied for 10 min. Ethyl isopropyl amiloride (EIPA) at 10 µM, 25 µM, 50 µM, and 100 µM was applied for 30 min. Bafilomycin A1 at 10 nM, 100 nM, and 500 nM was applied for 30 min. Phenylarsine oxide at 1 µM, 5 µM, 10 µM, and 100 µM was applied for 30 min. After respective inhibitor treatment, 1 µM of fluorescent paclitaxel was added for 30 min. Cells were then rinsed with PBS before 1:200 Hoechst for nuclear labeling was added for 10 min at 37 °C. Fluorescent paclitaxel and Hoechst’s fluorescence intensity were measured by using the plate reader at 540 nm and 465 nm, respectively.

To investigate paclitaxel transport specificity, cell experiments with two paclitaxel derivatives, namely docetaxel and cephalomannine, were performed. Here, 2.5 × 104 F11 cells were seeded in black 96-well plates and cultured in phenol red-free medium for 48 h. Afterward, cells were treated with CreEL paclitaxel, docetaxel, or cephalomannine at 0.1 µM, 0.5 µM, 1 µM, and 10 µM for 24 h. Furthermore, 1 µM fluorescent paclitaxel was added for 30 min. Cells were then rinsed with PBS before 1:200 Hoechst was added for 10 min at 37 °C. Fluorescent paclitaxel and Hoechst’s fluorescence intensity were measured by using the FLUOstart Omega plate reader (BMG LABTECH, Ortenberg, Germany) at 540 nm and 465 nm, respectively.

To investigate drug transport in paclitaxel translocation, different drugs were used to inhibit OATPs (Table 3). Here, 2.5 × 104 cells per well were seeded in a black 96-well plate and cultured in phenol red-free medium for 48 h. The drugs used as inhibitors were dissolved in a 100 mM (respective for all drugs dissolved in DMSO or NaOH) or 10 mM (respective for all drugs dissolved in H_2_O) stock solution (Table 3). Drugs were administered at increasing concentrations (0.0064 µM, 0.032 µM, 0.16 µM, 0.8 µM, 4 µM, 20 µM, 100 µM, and 500 µM) in 10% FBS phenol red-free DMEM medium and incubated for 2 h. After incubation, the medium was replaced with 1 mM fluorescent paclitaxel in 10% FBS phenol red-free medium. After 30 min, the medium was replaced with PBS and fluorescence was measured with the plate reader at 540 nm. Cells were incubated with 1:200 Hoechst in PBS for 10 min. The medium was replaced again with PBS, and fluorescence intensity was measured at 540 nm. Blanks and controls were provided.

#### 2.1.4. Overexpression of Drug Transporters

To obtain correct inserts for overexpression, different primers targeting the desired gene were used (Appendix A). Depending on the gene, different human cDNA templates, such as liver, kidney, brain, or ovarian, were used to amplify the respective genes. PCR products were digested with the appropriate restriction enzymes and cloned into a modified sleeping beauty transposon expression vector. Expression constructs were transfected into F11 cells. Cells were selected with puromycin and subsequently expanded in triple flasks. Expression of the desired transporter was induced via doxycycline. All overexpressing cells were tested for target protein expression via immunofluorescence staining against human influenza hemagglutinin (HA), as all successfully transfected cells expressed HA.

#### 2.1.5. Knock-Out of Drug Transporters

CRISPR/Cas9 knock-out cell lines of OATP1A1 and OATP1B2 were generated to further investigate drug-transporter involvement in paclitaxel-caused neurotoxicity (Appendix A). Thus, 2 × 105 F11 cells were seeded into 6-well plates. When cells were 70–90% confluent, transfection with the lipofectamine^TM^ 3000 transfection kit was performed, following the supplier’s instructions. Here, 5 µL Cas9 protein and 2 µL gRNA were used. Transfected cells were sorted and single clones analyzed for successful knock-out via RT-qPCR.

### 2.2. MTT Assay

A 3-(4,5-Dimethylthiazol-2-yl)-2,5-diphenyltetrazoliumbromid (MTT) assay was performed (i) to determine whether inhibitor pretreatment shows neurotoxic effects; (ii) to assess the neuroprotective effects of transporter knock-out; and (iii) to investigate whether inhibitor treatment sustains the neoplastic characteristics of paclitaxel. To accomplish these objectives, 5 × 103 cells per well were seeded onto a 96-well plate and cultured for 48 h. Cells were incubated for 2 h with inhibitors (Appendix A) before media were replaced with increasing paclitaxel concentrations containing phenol red-free medium. Cells were incubated for 72 h, and MTT Assay (M2128, Sigma-Aldrich, St.Louis, MO, USA) was performed following supplier’s instructions. Absorption was measured in the plate reader at 570 nm against the background of 690 nm.

### 2.3. RT-qPCR

Following the supplier’s instructions, RNA was isolated by using the MasterPureTM Complete DNA & RNA Purification Kit (Lucigen, Hoddesdon, UK). RNA solution was stored at −20 °C until further use.

To synthesize the cDNA of the RNA of all samples, the QuantiTect Reverse Transcription Kit (Qiagen, Hilden, Germany) was used according to the supplier’s instructions. All RT-qPCR experiments were performed in 96-well optical plates with triplets of each sample. Primer annealing temperatures for drug-transport primers (Appendix A) and neurotoxicity primers (Appendix A) were adapted to the melting temperature of the used primer pair. Results were normalized to GAPDH.

### 2.4. LC-MS/MS

To obtain samples for the following LC-MS/MS measurements, 5 × 104 overexpressing, knock-out, or F11 cells were seeded into a 24-well plate. Overexpressing and knock-out cells were treated with 0.78 µM, 3.125 µM, or 12.5 µM of paclitaxel for 15 min. For paclitaxel kinetics, cells were treated as described above for 15 min. For the active transport inhibition experiments, cells were treated with increasing inhibitory drug concentrations for 1 h and subsequently for 15 min with 3.125 µM of CreEL paclitaxel. After each treatment, the medium was removed, and cells were washed with ice-cold PBS. To lyse cells, 200 µL of RIPA buffer was added, and cells were incubated for 45 min on ice. Aliquots of the acquired lysates were used for LC-MS/MS paclitaxel concentration measurement and BCA protein assay. Cell lysates were further homogenized via ultrasound sonification for the BCA protein assay (see below).

For the LC-MS/MS measurements, 50 μL cell lysate was mixed with 50 μL of acetonitrile and 50 μL of acetonitrile solution containing stable isotope-labeled paclitaxel ([^13^C_6_]-paclitaxel), serving as the internal standard. Samples were centrifuged for 10 min at 12,000 rpm. Supernatants were removed and subjected to LC-MS/MS, as described before [17].

Paclitaxel concentration of each sample needed to be normalized with the number of cells that it derived from. BCA protein assay (23,250, Thermo Fisher Scientific, Waltham, MA, USA) was performed to define protein amounts in the cell lysate sample. Supplier’s instructions were followed; absorbance was measured in a plate reader at 560 nm; and the protein concentrations of the samples were calculated by using the BSA reference absorbance. All results of LC-MS/MS were depicted as ng paclitaxel per µg cells (*y*-axis) correlated with the different concentrations that were applied to the cells (*x*-axis).

### 2.5. Isolation and Culture of Murine Primary Sensory Neurons

Primary sensory neurons were obtained from 8–12-week-old male wild-type C57BL/6 mice. DRGs were isolated, attached nerves removed, and the DRGs transferred into 2.5 mL cell culture medium containing collagenase. For digestion, DRGs were placed in a sterile incubator with 95% air and 5% CO_2_ at 37 °C for 1 h. Afterward, DRGs were dissociated by triturating, and the cell suspension was centrifuged with a 15% BSA gradient for 8 min at 120× *g*. For paclitaxel uptake experiments, the obtained cells of one animal were used for one experiment and were thus plated accordingly on laminin-/ornithine-coated wells. Cells were cultured for 24 h in Neurobasal-A medium with 2% B-27 supplement 50×, 0.5 mM L-glutamine, 25 µM L-glutamate, and 1% penicillin-streptomycin.

### 2.6. Immunofluorescence and Microscopy

#### 2.6.1. Immunofluorescence Staining of Cells

After treatment, cells were fixed for 10 min with 4% PFA. Cells were washed three times with PBS. Cells were treated for 1 h at room temperature with normal goat serum blocking solution (2% goat serum, 1% bovine serum albumin (BSA), 0.1% Triton X-100, 0.05% Tween 20 in PBS). Antibody anti-UCHL1 (Abcam Cat# ab72910, RRID:AB_1269734) was diluted in 1:200 in 1% BSA and incubated at 4 °C overnight. Cells were again washed with PBS before DAPI in 1:100; the second antibody, AlexaFluor647 (Thermo Fisher Scientific Cat# A-21449, RRID:AB_2535866), in 1:1000 was added; and cells were incubated for 1 h at room temperature. Cells were washed again, and for analysis, 100 μL of PBS per well was added.

#### 2.6.2. Quantitative Microscopy

We used a Cellomics ArrayScan XTI microscope with an LED light source to acquire images of stained DRG cultures in 96-well imaging plates (Greiner µclear). Binned (2 × 2) images (1104 × 1104 pixels) were acquired with a 10× (NA = 0.3) EC Plan Neo-Fluor objective (Zeiss, Oberkochen, Germany) and analyzed using the Cellomics software package. Images of UCHL1-labeled neurons were background corrected (low pass filtration), converted to binary image masks (fixed threshold), and segmented (geometric method), and neurons were identified by using the object selection parameters: size of 80–7500 µm^2^, circularity (perimeter2/4π area) of 1–3, length-to-width ratio of 1–2, average intensity of 800–12,000, and total intensity of 2 × 10^5^–5 × 10^7^. The image masks were then used to quantify the signal of fluorescence-labeled paclitaxel.

### 2.7. Animals

In this study, 8–12-week-old female and male wild-type C57BL/6 mice (IMSR Cat# JAX_000664,RRID:IMSR_JAX:000664) were used. Mice were maintained on a 12 h light–dark cycle, held on sawdust bedding in plastic cages with food and water provided ad libitum. Animal experiments conformed with local state authorities (Landesamt für Natur, Umwelt und Verbraucherschutz Nordrhein-Westfalen). In total, 43 animals (23 male, 20 female) were used. Three mice were used for primary sensory cell culture, and 40 mice were used to study neurotoxicity in vivo, comparing a paclitaxel treatment group with a glycyrrhizic acid plus paclitaxel treatment group.

#### 2.7.1. Drug Administration

To study paclitaxel’s neurotoxic effects and their possible prevention via transport inhibition, 25 mg/kg of CreEL paclitaxel was injected intraperitoneally on d0 and d7. The control group received CreEL paclitaxel injections only, while another treatment group intraperitoneally received an injection of 100 mg/kg of glycyrrhizic acid 3 h prior to CreEL paclitaxel injection. CreEL paclitaxel was obtained via the University Hospital Pharmacy. The glycyrrhizic acid powder was purchased (Sigma-Aldrich, St.Louis, MO, USA) and resubstituted in H_2_O to a total of 100 mM. The general clinical condition was monitored daily, and body weight was assessed on d0 and d7. After treatment, mice were sacrificed by cervical dislocation.

#### 2.7.2. Nerve Conduction Studies

Nerve conduction studies of compound motor action potential (CMAP) and sensory nerve action potential (SNAP) were performed blinded on d0 and d11. Recordings of compound muscle action potential (CMAP) amplitudes, sensory nerve action potential (SNAP) amplitudes, and latencies were performed on a PowerLab single acquisition setup (ADInstruments, Grand Junction, CO, USA), as described before [26].

### 2.8. Statistical Analysis

All data were randomly collected and blindly assessed. For statistical analyses, the software programs MS Excel (Microsoft, Redmond, Washington, USA) and Prism 5.0 (GraphPad, San Diego, CA, USA) were used. Significances were analyzed between different groups by using the Mann–Whitney U test to compare two groups and one-way nonparametric ANOVA (Kruskal–Wallis test) for comparisons between more than two groups. If there were more than two groups that needed to be compared, an additional Dunn’s multiple comparison test was performed. Differences between groups were considered statistically significant when the *p*-value was at or below 0.05. Four levels of significance are highlighted: * *p* < 0.05, ** *p* < 0.01, *** *p* < 0.001, **** *p* < 0.0001.

## 3. Results

### 3.1. Neuronal Cell Line Rapidly Takes up Paclitaxel via an Active Transport Mechanism

Recent studies indicate that paclitaxel is taken up into neurons via active transport mediated by different OATPs, particularly OATP1B2 [22]. However, alternative passive transport was not yet excluded. Therefore, we first analyzed possible passive paclitaxel uptake into the neuronal cell model F11.

To detect possible passive transport mechanisms, we preincubated F11 cells with inhibitors of phagocytosis (phenylarsine oxide at 1 µM, 5 µM, 10 µM, and 100 µM), caveolae-mediated endocytosis (indomethacin at 75 µM, 150 µM, 225 µM, and 300 µM), micropinocytosis (EIPA at 10 µM, 25 µM, 50 µM, and 100 µM), clathrin-mediated endocytosis (chlorpromazine at 0.0064 µM, 0.016 µM, 4 µM, and 100 µM), and autophagy (bafilomycin A1 at 10 nM, 100 nM, and 500 nM) before exposure to 1 µM of paclitaxel. Paclitaxel uptake was not altered by any inhibitor treatment with either phenylarsine oxide, indomethacin, EIPA, or bafilomycin (Figure 1a). Inhibiting clathrin-mediated endocytosis with chlorpromazine showed significant effects only at 0.0064 µM and 0.016 µM.

Given the intracellular accumulation, paclitaxel concentrations rapidly increased over 0.5–2.0 h after exposure to different concentrations of fluorescent paclitaxel (Figure 1b). This accumulation seems to be saturable. F11 cells exposed to paclitaxel for up to 24 h demonstrate a rapid increase within the first hour, reaching a plateau after 2 hours (Figure 1c).

We subsequently assessed whether the transport of paclitaxel is specific to paclitaxel itself. F11 cells were therefore preincubated for 24 h with paclitaxel or structurally related derivatives docetaxel and cephalomannine (at 0.1 µM, 0.5 µM, 1 µM, and 10 µM) before 1 µM of paclitaxel exposure. Preincubation with paclitaxel and docetaxel, but not cephalomannine, significantly decreased paclitaxel uptake (Figure 1d). Preincubation with paclitaxel and docetaxel inhibited subsequent paclitaxel uptake in a dose-dependent fashion (Figure 1e). These findings exclude the passive transport of paclitaxel into the neuronal cell model and suggest the existence of a specific transport mechanism for paclitaxel.

### 3.2. Paclitaxel Is Actively Transported by OATP1A1, OATP1B2, and OATP2A1 in a Neuronal Cell Line

Our previously described findings suggest a drug-transport mechanism for paclitaxel. Hence, we examined the expression of OATPs in rat DRG tissue, rat brain tissue, neuronal cell line F11, and pancreatic cancer cell line AR42J (Figure 2a). Compared with rat DRG tissue, OATP1A1, OATP1A4, OATP1A5, and OATP1B2 are decreasingly expressed, while OATP2A1 and OATP2B1 are increasingly expressed in rat brain. Except for OATP1A1, all OATPs were more highly expressed in F11 or AR42J cells compared with DRG tissue.

To determine whether paclitaxel is actively transported by one of these transporters, we incubated neurons with known pharmacological inhibitors at increasing concentrations. Neuronal cell line F11 was incubated with drug-transport inhibitors (Table 3) at 0.0064 µM, 0.032 µM, 0.16 µM, 0.8 µM, 4 µM, 20 µM, 100 µM, or 500 µM. After 1 h of incubation with drug-transport inhibitors, cells were exposed to either 1 µM fluorescent paclitaxel with fluorescence measured on the plate reader (Figure 2b) or 3 µM of solvent-based paclitaxel with concentrations measured by LC-MS/MS (Figure 2c). Glycyrrhizic acid, taurocholate, ibuprofen, and niflumic acid (OATP1A1, OATP1A4, OATP1B2, OATP2A1, and OATP2B1) inhibited fluorescent paclitaxel uptake by about 25% (Figure 2b). Solvent-based paclitaxel uptake was inhibited by about 50% with taurocholate, ibuprofen, and naringin (OATP1A1, OATP1A4, OATP1A1, OATP1B2, and OATP2B1). OATP2A1 inhibition via niflumic acid decreased solvent-based paclitaxel translocation into the neuron by approximately 60%. However, the most potent drug was glycyrrhizic acid, which reduced paclitaxel uptake by about 75% at a concentration of 100 µM (Figure 2c). Hence, paclitaxel is most likely actively transported by OATP1A1, OATP1B2, and OATP2A1.

### 3.3. Overexpression of OATPs Increases Paclitaxel Uptake

On the basis of the expression profile of the described drug transporters and inhibitor experiments, we conclude that OATP1A1, OATP1B2, and OATP2A1 are the most potent candidates for paclitaxel transport. OATP1A1 and OAT1B2 in rodents correspond to OATP1A2 and OATP1B1 in humans, respectively. OATP2A1 is both expressed in humans and rodents. Those transporters were subsequently overexpressed in F11 cells to verify if paclitaxel uptake is altered (Figure 3).

An increased intracellular concentration of paclitaxel via LC-MS/MS was observed after the overexpression of candidate transporters OATP1A1 (rodent), OATP1A2 (human), OATP1B1 (human), and OATP1B2 (rodent) (Figure 3a–d). No changes in paclitaxel uptake were observed in OATP2A1 overexpressing cell lines (Figure 3e).

### 3.4. Primary Murine DRG Neurons Actively Take up Paclitaxel

Subsequently, we analyzed the uptake of paclitaxel into primary sensory neurons isolated from the dorsal root ganglia (DRG) of adult mice. For this, we applied a high content screening (HCS) microscopy-based approach that provides single-cell data of DRG neurons and non-neuronal cells [26,27]. For single-cell analysis, we stained murine neurons with a neuronal marker, Ubiquitin carboxy-terminal hydrolase L1 (UCHL1), to identify the neurons (Figure 4a). We observed a reduced number of analyzable neurons at the highest concentration, 10 µM of paclitaxel, which was probably due to the increased fragility of the cells as a result of the treatment (Figure 4b). Similar to F11 cells, we observed a rapid and dose-dependent uptake (Figure 4c). Paclitaxel uptake could be significantly reduced when primary murine cell cultures were treated with 250 µM of glycyrrhizic acid (Figure 4d). We found that paclitaxel is taken up into distinct subpopulations of DRG neurons, as not all UCHL1-positive cells are paclitaxel positive (Figure 4e). Figure 4a moreover depicts a UCHL1-positive neuronal cell that did not accumulate paclitaxel.

### 3.5. Glycyrrhizic Acid Inhibits OATP1A1- and OATP1B2-Mediated Paclitaxel Transport

Because glycyrrhizic acid was the most potent inhibitor of paclitaxel uptake, we next used the CRISPR/Cas9-System to knock out the glycyrrhizic-acid-susceptible transporters, OATP1B2 and OATP1A1, in neuronal F11 cells (Figure 5). Treatment of F11 knock-out cells with 1 µM of paclitaxel resulted in significantly decreased paclitaxel accumulation in the OAPT1B2- and OATP1A1-deficient cell lines, indicating that these are specific paclitaxel transporters (Figure 5a). Furthermore, the knock-out of transporters lead to significantly increased cell viability, deducted by the MTT assay, after exposure to 1 µM of paclitaxel for 72 h when compared with vehicle knock-out cells (Figure 5b).

### 3.6. OATP Inhibition Protects against Paclitaxel-Induced Neurotoxicity In Vitro

MTT assays were performed to assess whether the inhibitor treatment is able to ameliorate the neurotoxicity of paclitaxel. Neuronal F11 cells were treated with increasing concentrations of inhibitory drugs, as described above, and 1 µM of paclitaxel for 24 h. No changes in cell viability were seen after treatment with taurocholate and ibuprofen. Cell viability further decreased after cotreatment with naringin and niflumic acid at higher concentrations. Treatment with glycyrrhizic acid drastically increased cell viability, to ~97%, at a concentration of 0.16 µM, indicating that higher concentrations of glycyrrhizic acid are able to rescue the cytotoxic effects of paclitaxel. As a positive control, neuronal F11 cells were treated with increasing concentrations of paclitaxel. After 24 h of treatment, paclitaxel concentrations higher than 20 µM resulted in 100% cytotoxicity (Figure 6a).

A wide array of toxicity markers was investigated via RT-qPCR to further study the prevention of paclitaxel-induced neurotoxicity. F11 cells were incubated with 1 µM of paclitaxel (Figure 6b) or with 10 µM of inhibitory drug and 1 µM of paclitaxel for 48 h (Figure 6c). Subsequently, RT-qPCR was performed to assess the expression level of anti-apoptotic marker BCL-2; proapoptotic marker BAX and BAD; apoptosis and cell cycle markers ATM, DAPK1, and USP7; neuroprotective markers GPR37 and BDNF; inflammatory markers NFKB1 and LTA; and actin filament marker GSN. We found that paclitaxel treatment on itself mainly increased the expression of GPR37 and NFKB1. Glycyrrhizic acid decreased BAX and BAD expression while increasing BCL-2 expression and further increased the expression of neuroprotective markers GPR37 and BDNF.

### 3.7. Inhibition of OATP Does Not Attenuate Paclitaxel Cancer Cell Toxicity

The potential interference of transporter modulation with antineoplastic properties of paclitaxel was tested in 11 cancer cell lines, namely A459, CCD-13Lu, CHO-K1, CHOwt, MCF-7, MDA-MB-231, MDA-MB-435, NCI-H460, PC3, and 4T1, which are known to be susceptible to paclitaxel. Cell lines were pretreated with inhibitors and then treated with increasing paclitaxel concentrations to explore a potentially increased resistance to paclitaxel (Figure 7).

To present data in a more comprehensive way, a color-coded heatmap of the results was created (Figure 7a). Red indicates absent toxicity to cancer cells with inhibitor plus paclitaxel treatment compared with paclitaxel treatment alone. Orange indicates reduced toxicity to cancer cells with inhibitor plus paclitaxel treatment. Yellow indicates no changes between treatments. Green indicates even enhanced toxicity to cancer cells with inhibitor plus paclitaxel treatment.

Only one treatment condition with glycyrrhizic acid in CHO-K1 cells led to no cancer cell death. Otherwise, all the other settings still were able to induce cancer cell death. Overall, the MTT assay results show that pretreatment with drug-transport inhibitors does not increase cancer cell survival (Figure 7b).

### 3.8. Glycyrrhizic Acid Prevents Paclitaxel Neurotoxicity In Vivo

We were able to inhibit the paclitaxel transport via OATP1A1 and OATP1B2 in immortalized neuronal cultures and primary sensory neuronal cultures. As glycyrrhizic acid is a natural drug that is safe in use and has an anti-inflammatory profile, it seems to be the perfect candidate for in vivo use. Mice were intraperitoneally injected with glycyrrhizic acid 3 h prior to paclitaxel treatment, consisting of intraperitoneal paclitaxel injections once a week for 2 weeks. Behavioral testing, neurography testing, and anatomical testing were performed (Figure 8). An acetone test was performed to assess the extent of paclitaxel-induced cold allodynia. The paclitaxel treatment significantly increased the reaction time. Glycyrrhizic acid was able to reverse paclitaxel-induced cold allodynia back to control levels (Figure 8a). Body weight (Figure 8b) and axon count (Figure 8g,h) did not change during treatment in either treatment group when compared with control animals. Compound motor action potentials (CMAP) significantly decreased with paclitaxel treatment, but not after combinatory treatment of glycyrrhizic acid and paclitaxel, when compared with control-treated animals (Figure 8c). Meanwhile, motor latency was not affected by different treatments (Figure 8e). Sensory nerve action potentials conducted at the caudal nerve of mice show significantly reduced amplitudes (Figure 8d), as well as latency (Figure 8f), after paclitaxel treatment, while cotreatment with glycyrrhizic acid reduced this effect to control levels. Additionally, compound motor action potentials (CMAP) significantly decreased with paclitaxel treatment, but not after combinatory treatment of glycyrrhizic acid and paclitaxel, when compared to control-treated animals (Figure 8c). Body weight (Figure 8d) and axon count (Figure 8e) did not change during treatment in either treatment group when compared to control animals.

## 4. Discussion

We in detail characterized the transport of paclitaxel into neurons and provided evidence that the inhibition of drug transporters efficiently attenuated the neurotoxicity of paclitaxel without interfering with its antineoplastic properties. In our initial experiments, we assessed a previously unaddressed issue: the possibility of an alternative, transporter-independent route of paclitaxel translocation. Three findings assured us that paclitaxel influx into neurons is a transporter-based process:Inhibition of endocytosis, autophagy, macropinocytosis, and clathrin- or caveolae-mediated transport did not attenuate paclitaxel translocation into neurons. An exception was increased paclitaxel uptake at high inhibitor concentrations, which was likely caused by increased inhibitor cytotoxicity at escalating concentrations. If cells undergo apoptosis and become porous, this could increase paclitaxel influx or the accumulation of fluorescence in the fluorescent paclitaxel that binds to cell structures.Our kinetic experiments demonstrate that peak paclitaxel concentrations in F11 were measured not before 5 h after incubation, which was much longer than one would expect in the case of uptake into the sensory neuron via diffusion. Diffusion is a highly efficient process, and equilibrium is reached after seconds or minutes.Pre-exposure of F11 cells with paclitaxel and its structurally closely related derivative docetaxel prevented paclitaxel uptake, which strongly indicates a saturable, specific transport mechanism. Cephalomannine, on the other hand, was not able to alter paclitaxel influx, probably because of the different hydroxylation sites in the molecular structure, and hence, its induced metabolic differences [28,29]. Fluorescent paclitaxel uptake is inversely correlated with the concentration of docetaxel and paclitaxel used. This indicates the competitive inhibition of fluorescent paclitaxel uptake. Notably, paclitaxel has been suggested to be a drug-transport inhibitor itself, inhibiting OATP1B2 and OATP2B1 [30,31]. This could further alter experiment outcomes with predispositions to paclitaxel and its derivatives.

Among the large number of drug transporters, most evidence indicates that the organic anion transporter polypeptide (OATP) family is critically involved in paclitaxel transport [21,22,32]. Of the examined OATP, we observed the most robust reduction in paclitaxel influx when OATP1A1 or OATP1B2 were either pharmacologically blocked by glycyrrhizic acid or genetically depleted by CRISPR/Cas9 knock-out. Furthermore, the overexpression of these transporters increased paclitaxel translocation. The relevance of transport inhibition for neurotoxic effects was further demonstrated by increased neuronal cell survival. OATP1B2 has already been recognized as a putative transporter for paclitaxel [22]. Our data, therefore, confirm the observations from the study by LeBlanc and colleagues: the blockage of OATP1B2 can partially prevent the neurotoxicity of paclitaxel. In contrast to LeBlanc et al., who used the FDA-approved tyrosine-kinase inhibitor nilotinib, we used the substance glycyrrhizic acid. Our data also provide an explanation for the only partial rescue of neurons by blockage with nilotinib as we identified at least one other relevant drug transporter, OATP1A1.

OATP1A1 is a transporter exclusively expressed in rodents, its human homolog being OATP1A2. In vivo studies on SLCO1a/1b^−/−^ 1A2^tg^ animals (SLCO1/ab knock-out, OATP1A2 transgenic mice), OATP1A2 has been described to transport paclitaxel, especially into the liver. The OATP1A2 transport of paclitaxel in transgenic mice was described as being higher than that in in OATP1B3 transgenic mice [33].

In our in vivo study, we used a well-established model of paclitaxel-induced neuropathy [13,34]. The functional test clearly demonstrates that small and larger fiber sensory neurons are dysfunctional and that an injection of glycyrrhizic acid prior to paclitaxel exposure prevented these pathological changes. We also observed reduced motor amplitudes. High doses of paclitaxel can, in addition to leading to abnormalities in sensory nerve conduction, lead to abnormalities in motor nerve conduction [35]. The rescue of motor amplitudes with glycyrrhizic acid shows that it has protective effects beyond primary sensory neurons. Changes in axon numbers were not observed with either treatment regime, as structural changes are expected to occur not before 6 weeks of paclitaxel treatment [36].

Glycyrrhizic acid is a naturally occurring component extracted from licorice plants [37,38]. It has anti-inflammatory [39], antiallergic [40], and antiviral properties [41] and has been propagated as a treatment for chronic hepatitis [42]. Glycyrrhizic acid has low toxicity and thus has lately been used as an oral absorption enhancer for hydrophobic drugs. In this setting, it can increase drug solubility, promote penetration through the cell membrane, and decrease cell membrane elasticity [43]. This micelle formation has also been studied in combination with paclitaxel, where it was able to increase the bioavailability of paclitaxel [44]. The micelle formation of paclitaxel and glycyrrhizic acid greatly increased the targeting of hepatic tumor cells and decreased tumor growth in mice [45].

Overall, we here provide evidence that OATP family members are able to translocate paclitaxel. We could modulate this transport to prevent paclitaxel uptake into the neuron and prevent neurotoxic effects while still sustaining the antineoplastic characteristics of paclitaxel (Figure 9). We have found OATP1A1 and OATP1B2 to be the most suitable targets for an in vivo study and have demonstrated that the inhibition of those transporters with glycyrrhizic acid is able to reverse induced peripheral neuropathy.

## Figures and Tables

**Figure 1 cells-12-01249-f001:**
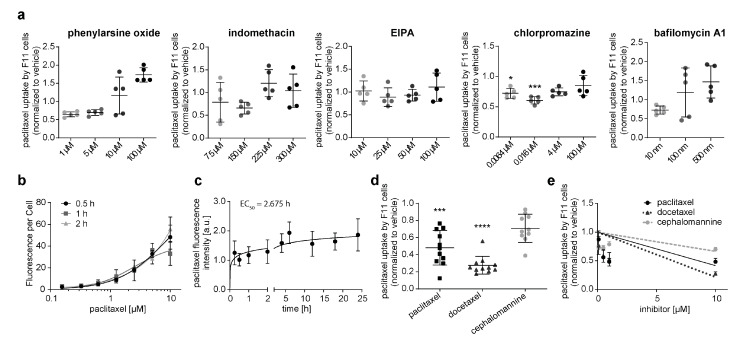
Kinetics and uptake mechanisms of paclitaxel and its derivatives (docetaxel and cephalomannine) in neuronal F11 cells. (**a**) F11 neuronal cells were treated for inhibitors of clathrin-mediated endocytosis via chlorpromazine, caveolae-mediated endocytosis via indomethacin, macropinocytosis via ethyl isopropyl amiloride (EIPA), and phagocytosis via bafilomycin A1. Subsequently, cells were treated with fluorescent paclitaxel and fluorescence measured via plate reader. (**b**) F11 cells were exposed to different concentrations of fluorescent paclitaxel (0.15 µM, 0.3 µM, 0.6 µM, 1.25 µM, 2.5 µM, 5 µM, and 10 µM), and fluorescence was measured via plate reader after 0.5 h, 1.0 h, or 2.0 h (*n* = 3). (**c**) Depicted is the uptake of 1 µM of fluorescent paclitaxel into F11 cells over 24 h (*n* = 12). (**d**) The neuronal cell line was preincubated for 24 h with paclitaxel, docetaxel, and cephalomannine before exposure to 1 µM of fluorescent paclitaxel (*n* = 10). (**e**) An increased concentration of paclitaxel and its derivatives leads to a linear decrease in paclitaxel uptake in F11 cells (*n* = 6). Graphs depict mean ± SD; *n* = independent cell culture preparations. Statistical comparison was performed between all groups (Kruskal–Wallis test, Dunn’s multiple comparison test); * *p* < 0.05, *** *p* < 0.001, **** *p* < 0.0001.

**Figure 2 cells-12-01249-f002:**
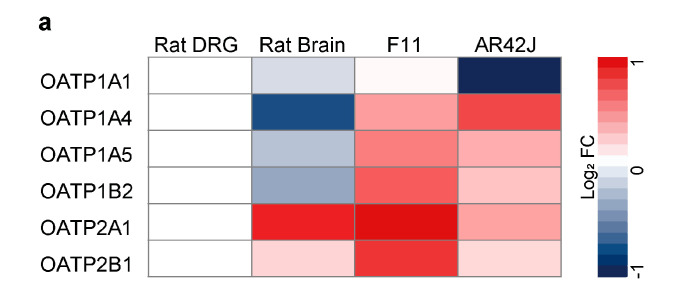
Inhibitors of OATP1A1, OATP1B2, and OATP2A1 prevent the transport of paclitaxel in F11 cells. (**a**) Heatmap depicts expression of different OATPs in rat brain, neuronal cell line F11, and pancreatic cancer cell line AR42J compared with rat DRG (*n* = 3). (**b**) F11 cells were treated with glycyrrhizic acid, taurocholate, ibuprofen, naringin, and niflumic acid for 2 h before treating them with 1 µM of fluorescent paclitaxel for 30 min. Cellular fluorescence intensity was measured and compared with vehicle-treated cells via plate reader (*n* = 8). (**c**) F11 cells were treated with glycyrrhizic acid, taurocholate, ibuprofen, naringin, and niflumic acid for 1 h before treatment with 3 µM of paclitaxel for 15 min. The paclitaxel amount in the sample was measured by LC-MS/MS, set in ratio with the amount of cell protein in the sample, and compared with vehicle-treated cells (*n* = 3). *n* = independent cell culture preparations.

**Figure 3 cells-12-01249-f003:**
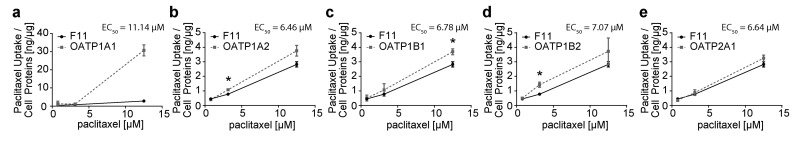
Overexpression of OATPs increases paclitaxel uptake in transfected F11 cells. Overview of F11 cell line and overexpressing cell lines exposed to 0.78 µM, 3.125 µM, and 12.5 µM of paclitaxel (*x*-axis) for 30 min. Graphs show paclitaxel amount in ng per µg cell. (**a**) OATP1A1 overexpressing cell line, (**b**) OATP1A2 overexpressing cell line, (**c**) OATP1B1 overexpressing cell line, (**d**) OATP1B2 overexpressing cell line, and (**e**) OATP2A1 overexpressing cell line paclitaxel uptake in comparison with F11 cell line paclitaxel uptake. *n* = 3 independent cell culture preparations. Graphs depict mean ± SD. Statistical comparison was performed between all groups (Kruskal–Wallis test, Dunn’s multiple comparison test); * *p* < 0.05.

**Figure 4 cells-12-01249-f004:**
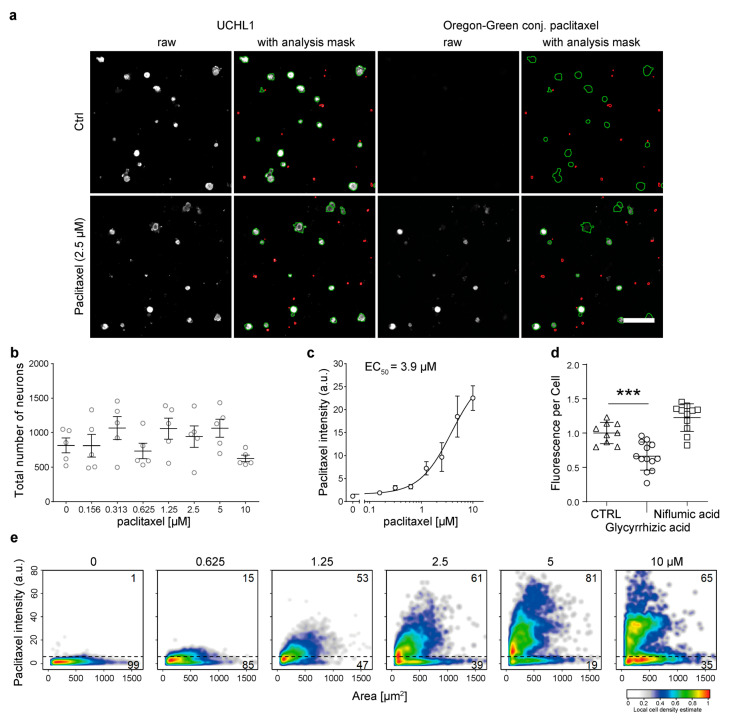
Paclitaxel kinetics in primary sensory neurons and impact of active transport inhibition on paclitaxel uptake. (**a**) Micrographs of primary sensory neurons in culture were stained for UCHL1 (red) and fluorescence in green thanks to paclitaxel uptake. Masking of cells indicates UCHL1-positive neuron that is paclitaxel negative. (**b**) Graph depicts mean cell number per well after paclitaxel treatment with increasing concentrations. (**c**) Graph displays mean paclitaxel intensity after paclitaxel treatment with increasing concentrations. (**d**) Primary sensory neurons were treated with glycyrrhizic acid and niflumic acid for 2 h before treatment with 1 µM of fluorescent paclitaxel for 30 min. Fluorescence intensity of cells was measured via plate reader and compared with vehicle-treated cells (*n* = 12). (**e**) Graphs depict the correlations between paclitaxel-positive cells and UCHL1-positive cells at increasing amounts of paclitaxel concentration. Graphs depict mean ± SD; *n* = independent cell culture preparations. Statistical comparison was performed between all groups (Kruskal–Wallis test, Dunn’s multiple comparison test); *** *p* < 0.001.

**Figure 5 cells-12-01249-f005:**
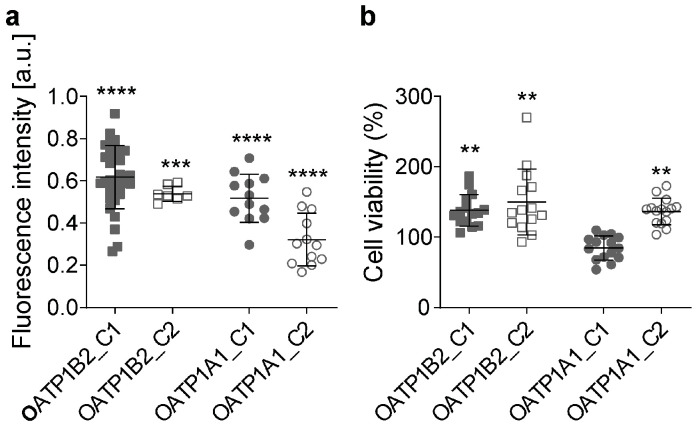
Influence of OATP1A1 and OATP1B2 knock-out on paclitaxel uptake into F11 cells. (**a**) OATP1A1-clone1 cell line (OATP1A1_C1), OATP1A1-clone2 cell line (OATP1A1_C2), OATP1B2-clone1 cell line (OATP1B2_C1), and OATP1B2-clone2 cell line (OATP1B2_C2) were treated with 1 µM of fluorescent paclitaxel for 30 min. Fluorescence intensity of cells was measured and compared with vehicle-treated cells (*n* = 8). (**b**) Knock-out cells were treated with 1 μM of paclitaxel for 72 h. Cell viability was assessed via MTT assay and compared with vehicle-treated cells (*n* = 15). Graphs depict mean ± SD; *n* = independent cell culture preparations. Statistical comparison was performed between all groups (Kruskal–Wallis test, Dunn’s multiple comparison test); ** *p* < 0.01, *** *p* < 0.001, **** *p* < 0.0001.

**Figure 6 cells-12-01249-f006:**
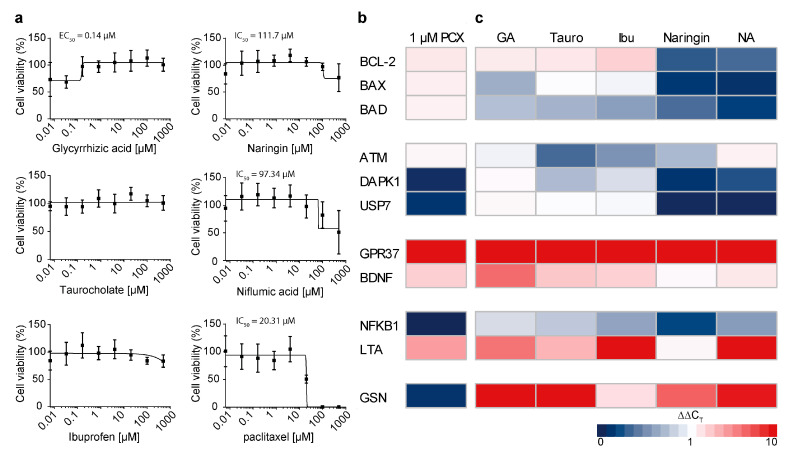
Prevention of neurotoxicity by OATP inhibition. (**a**) F11 cells were treated with glycyrrhizic acid, naringin, taurocholate, niflumic acid, and ibuprofen and with 1 µM of paclitaxel for 24 h. Control F11 cells were treated with increasing concentrations of paclitaxel for 24 h. Cell viability was assessed via MTT assay by comparing treated cells with vehicle-treated cells (*n* = 4). Graphs depict mean ± SD. (**b**) F11 cells were treated with 1 µM of paclitaxel for 48 h, and the expression of markers was compared with vehicle-treated F11 cells. (**c**) F11 cells were treated with 1 µM of paclitaxel and drug-transport inhibitors for 48 h, and the expression of markers was compared with F11 cells treated with only 1 µM of paclitaxel (*n* = 3). Graphs depict mean ± SD; *n* = independent cell culture preparations.

**Figure 7 cells-12-01249-f007:**
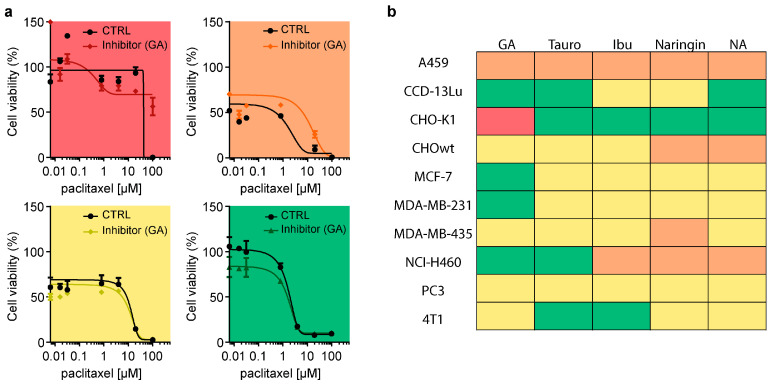
Viability changes of cancer cells after treatment with OATP inhibitors and paclitaxel. (**a**) Representative results of MTT cell viability assay. Red (represented with CHO-K1 cell line with glycyrrhizic acid (GA)) indicates complete loss, and orange (A459 cell line with GA) indicates significant loss of antineoplastic properties with inhibitor plus paclitaxel treatment compared with paclitaxel treatment alone. Yellow (MDA-MB-435 cell line with GA) indicates no changes between treatments. Green (MDA-MB-231 cell line with GA) indicates enhanced toxicity to cancer cells with inhibitor plus paclitaxel treatment compared with paclitaxel treatment alone. (**b**) Cancer cell lines A459, CCD-13Lu, CHO-K1, CHOwt MCF-7, MDA-MB-231, MDA-MB-435, NCI-H460, PC3, and 4T1 were treated with inhibitors, namely glycyrrhizic acid (GA), taurocholate (Tauro), ibuprofen (Ibu), naringin, and niflumic acid (NA), plus increasing concentrations of paclitaxel for 72 h. Cell viability was compared with paclitaxel-treated cells, and the results were indicated as described in (**a**) in the corresponding color (*n* = 3). Graphs depict mean ± SD; *n* = independent cell culture preparations.

**Figure 8 cells-12-01249-f008:**
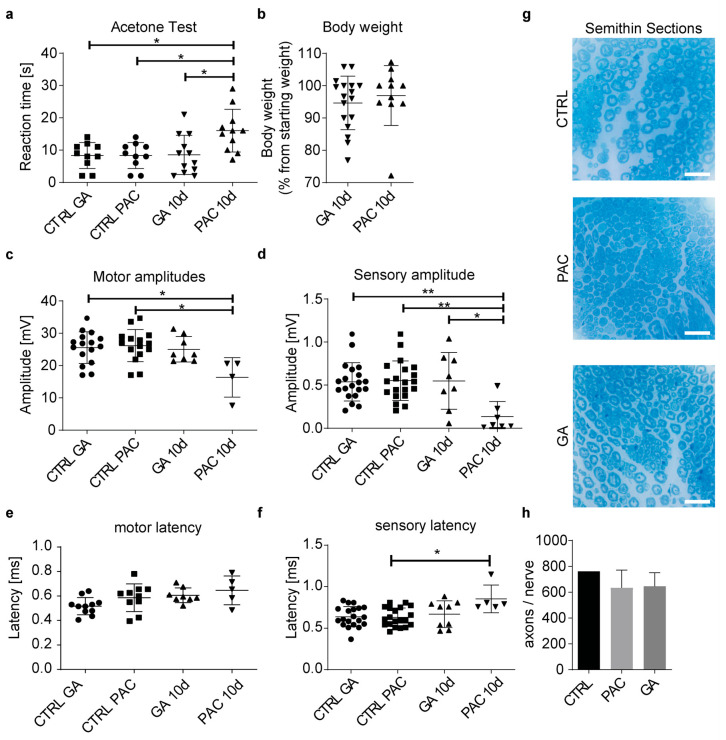
Influence of glycyrrhizic acid on paclitaxel-induced peripheral neuropathy in vivo. Animals were treated with paclitaxel (PAC) or glycyrrhizic acid plus paclitaxel (GA) two times, on d0 and d7. Tests were performed on d0 and d11. Control (CTRL) values are d0 values. (**a**) Behavioral acetone test reveals significantly reduced reaction time in paclitaxel-treated group compared with the GA-plus-paclitaxel-treated group (CTRL GA: *n*= 10, CTRL PAC: *n* = 10, GA: *n* = 12, PAC: *n* = 12). (**b**) Animals were weighed daily, and weight was compared to d0. No changes were observed (GA: *n* = 17, PAC: *n* = 10). (**c**) Motor amplitudes conducted at the sciatic nerve show decreased amplitude in the paclitaxel-treated animals compared to control animals (CTRL GA: *n*= 16, CTRL PAC: *n* = 15, GA: *n* = 8, PAC: *n* = 5). (**d**) Sensory nerve action potentials were performed at the caudal nerve of the mice. Reaction time amplitudes were significantly reduced in the paclitaxel-treated group compared with the GA-plus-paclitaxel-treated group (CTRL GA: *n*= 20, CTRL PAC: *n* = 20, GA: *n* = 8, PAC: *n* = 7). (**e**) Motor latencies between the groups showed no differences (CTRL GA: *n*= 11, CTRL PAC: *n* = 10, GA: *n* = 8, PAC: *n* = 5). (**f**) Sensory latency is significantly increased in the paclitaxel treatment group at d10 when compared with the CTRL group (CTRL GA: *n*= 20, CTRL PAC: *n* = 20, GA: *n* = 9, PAC: *n* = 5). (**g**) Micrographs depict semithin sections of different treatment groups: CTRL, paclitaxel treatment group, GA treatment group. (**h**) Axons were counted per examined nerve. Graphs depict mean ± SD; scale bar = 100 µm; *n* = number of animals. Statistical comparison was performed between all groups (Kruskal–Wallis test, Dunn’s multiple comparison test); * *p* < 0.05, ** *p* < 0.01.

**Figure 9 cells-12-01249-f009:**
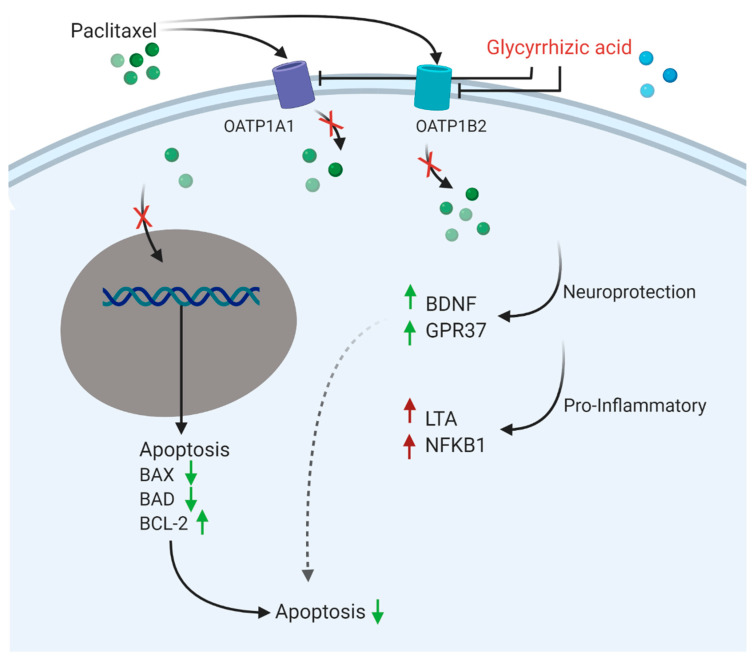
Diagram of possible effects of glycyrrhizic acid.

**Table 1 cells-12-01249-t001:** Cell lines and used cell culture medium.

Cell Line	Medium	FBS Concentration
F11; A549; CHO-K1; CHOwt; PC-3	DMEM	10%
AR42J	F12-K	20%
4T1; NCI-H460	RPMI-1640	10%
CCD-13Lu; MCF-7	EMEM	10%
MDA-MB-231; MDA-MB-435	L15	10%

**Table 2 cells-12-01249-t002:** Respective drug to inhibit uptake mechanism.

Uptake Mechanism	Inhibitory Drug
Clathrin-mediated endocytosis	Chlorpromazine
Caveolae-mediated endocytosis	Indomethacin
Macropinocytosis	Ethyl isopropyl amiloride (EIPA)
Autophagy	Bafilomycin A1
Phagocytosis	Phenylarsine oxide
Uptake mechanism	Inhibitory drug

**Table 3 cells-12-01249-t003:** Respective inhibitory drug, transporter target, and solvent.

Inhibitor	Transporter	Solvent
Ibuprofen	OATP1A1	DMSO
	OATP2B1	
Glycyrrhizic acid	OATP1A1	H_2_O
	OATP1B2	
Taurocholate	OATP1A4	H_2_O
	OATP1B2	
Naringin	OATP1A2	DMSO
	OATP1A5	
	OATP2B1	
Niflumic acid	OATP2A1	DMSO
Ibuprofen	OATP1A1	DMSO
	OATP2B1	

## Data Availability

Data are available upon request.

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
