# Peer review of "Glycyrrhizic Acid Prevents Paclitaxel-Induced Neuropathy via Inhibition of OATP-Mediated Neuronal Uptake"

_cells, 2023, doi:10.3390/cells12091249_

Round 1

Reviewer 1 Report

This manuscript explores the use of glycyrrhizic acid in preventing paclitaxel-induced neuropathy. Glycyrrhizic acid inhibits OATPs, reducing paclitaxel intake and alleviating paclitaxel-associated toxicity. Thematically, this study is well suited for the special issue of Cells but suffers from some unclear experiment design at times as well as some other concerns:

1.      Fig. 4d needs vehicle treatment as the control.

2.      Western blots confirming the KO of OATP1B2 and OATP1A1 should be included in the supplementary figures. Apologies if I missed it, there are no supplementary files included with the manuscript and the link for supplementary materials doesn’t work.

3.      Fig. 6 was generated using a paclitaxel dose of 1 µM, which seems like a low dose from Fig. 4c and 6a bottom right. 1 µM paclitaxel has no impact on cell viability. A higher dose near the middle of the slope (like 5 µM) might be better.

4.      Why does the lowest dose of glycyrrhizic acid decrease viability if 1 µM paclitaxel has no impact on cell viability? Cell viability is near 100% for all four other assessed treatments.

5.      In general there doesn’t seem to be a good anti-correlation between paclitaxel uptake in Fig. 2b-c and cell viability in Fig 6a. Can the authors explain why?

6.      What is the expression of OATPs on the cancer cell lines in Fig. 7?

7.      In Fig. 7a, please label or state in figure captions which conditions the representative results came from. Please include the remaining dose-response curves in the supplement

8.      In Fig. 7, why were the cells pretreated with OATP inhibitors before Paclitaxel treatment, rather than concurrent treatment?

9.      Why wasn’t the in vivo experiment performed on tumor-bearing mice? Without tumors established from a cell line used in Fig. 7, it cannot be determined if glycyrrhizic acid treatment impacts paclitaxel efficacy and validate the results of Fig. 7 in vivo.

10.    Statistical analysis is missing from many figures like the majority of Fig. 1, 3, 4b, and 6. IC50 values should be reported in all dose-response curves (Fig. 2, 6, and 7).

11.    Sentence in line 498 is cut off before completion.

Author Response

  1. 4d needs vehicle treatment as the control.

We thank the reviewer for this comment and added the vehicle/control treatment.

  1. Western blots confirming the KO of OATP1B2 and OATP1A1 should be included in the supplementary figures. Apologies if I missed it, there are no supplementary files included with the manuscript and the link for supplementary materials doesn’t work.

This is an interesting comment as we also tried to implement this in this study. Due to the unavailability of OATP1B2 antibodies for Western Blot and the inability to establish OATP1A1 Western Blot in our lab, we cannot repeat these experiments at the moment. Until the necessary tools and techniques become available, we are limited in our ability to perform confirmation western blots.

  1. Fig. 6 was generated using a paclitaxel dose of 1 µM, which seems like a low dose from Fig. 4c and 6a bottom right. 1 µM paclitaxel has no impact on cell viability. A higher dose near the middle of the slope (like 5 µM) might be better.

We appreciate the reviewer's comment. When compared to the cumulative dose of treated cancer patients (1,35-3,75 mg/m²) the corresponding concentration in cell culture would be indeed laying between 1-5 µM. In our study, we aimed to maintain consistency with previous research that has measured effects of paclitaxel at lower concentrations in the nanomolar range. Even at 0.5 µM, we observed significant changes in cell nuclei size in previously published data. Therefore, as we also expose cells to an elongated time of paclitaxel, using a higher concentration of paclitaxel would not align with our objectives of maintaining uniformity in our data analysis.

  1. Why does the lowest dose of glycyrrhizic acid decrease viability if 1 µM paclitaxel has no impact on cell viability? Cell viability is near 100% for all four other assessed treatments.

It is very interesting that the reviewer is pointing that out as we have expected something else as well. Paclitaxel cell viability decreased to 84% with 1 µM Paclitaxel, while it decreases to 73% and 69% when treated with low concentrations of glycyrrhizic acid. We would have expected cell viability to decrease with increasing concentrations of inhibitor. This could mean one of three things, low doses of glycyrrhizic acid are toxic to neurons, low doses of glycyrrhizic acid in combination with paclitaxel are neurotoxic, or glycyrrhizic acid is indeed able to decrease and prevent paclitaxel uptake into the cells and this effect only sets in with higher concentrations of glycyrrhizic acid. We agree, that the differences between the cell viabilities does not vary much. As such this should be seen as an observed trend.

  1. In general there doesn’t seem to be a good anti-correlation between paclitaxel uptake in Fig. 2b-c and cell viability in Fig 6a. Can the authors explain why?

We are grateful to the reviewer for their thoughtful question. Our analysis in figure 2b indicates that there is little change in neuronal numbers. We utilized UCHL-1 as a neuronal marker but this does not distinguish between damaged and healthy neurons. As shown in figure 2C, we observe the uptake of paclitaxel into neurons, and in figure 6a, we show the impact of paclitaxel on cell viability, proliferation, and cytotoxicity in neuronal cells. We believe that these two data sets are strongly correlated. For instance, we observe that with increasing paclitaxel concentration, the neurons take up more paclitaxel (as shown in figure 2C), which in turn has toxic effects on neuronal cells (as displayed in figure 6a). It is also important to note that apoptotic neuronal cells have a tendency to detach from plastic/coating of the cell culture flasks. During washing steps in experiments these apoptotic cells are most likely removed. Consequently, any measured cytotoxic effects may be reduced.

  1. What is the expression of OATPs on the cancer cell lines in Fig. 7?

The expression of OATPs on multiple human cancer cell lines has been studied before, like A549, MCF-7, MDA-MB-231, and MDA-MB-468. However, the expression of non-human cancer cell lines, that were used in this study as well, are not well examined. Unfortunately, we did not further investigate the expression of OATPs in those cancer cell lines. However, we only used cancer entities that are known to express OATPs, since the expression pattern of OATPs on different cancer entitites is well studied. The paper of Obaidat et al., 2011 gives a great overview of the tissue expression of OATPs in cancer cells, from which we designed our experiments.

  1. In Fig. 7a, please label or state in figure captions which conditions the representative results came from. Please include the remaining dose-response curves in the supplement

We kindly thank the reviewer for the suggestion and added the corresponding cell line and treatment to the figure legend, as well as the remaining curves in the supplementary figures. “Figure 7. Viability changes of cancer cells after treatment with OATP inhibitors and paclitaxel. (a) Representative results of MTT cell viability assay. Red (represented with CHO-K1 cell line with glycyrrhizic acid (GA)) indicates complete loss, orange (A459 cell line with GA) significant loss of antineoplastic properties with inhibitor plus paclitaxel treatment compared to paclitaxel treatment alone. Yellow (MDA-MB-435 cell line with GA) indicates no changes between treatments. Green (MDA-MB-231 cell line with GA) indicates enhanced toxicity to cancer cells with inhibitor plus paclitaxel treatment compared to paclitaxel treatment alone.”

  1. In Fig. 7, why were the cells pretreated with OATP inhibitors before Paclitaxel treatment, rather than concurrent treatment?

Here we followed protocol for established and published protocol by Leblanc et al., 2018. Moreover, we repeated those experiments with different incubation times as well as concurrent treatment, which we can provide if necessary. However, for comparability to other studies we chose to include this experimental setup.

  1. Why wasn’t the in vivo experiment performed on tumor-bearing mice? Without tumors established from a cell line used in Fig. 7, it cannot be determined if glycyrrhizic acid treatment impacts paclitaxel efficacy and validate the results of Fig. 7 in vivo.

This is an outstanding regard of the reviewer. Indeed, in a follow-up multi-central experimental setup we repeated OATP-inhibition with tumor-bearing mice. These data are not published yet. However, should be later this year. Figure 7 should underline the possibility to use OATP-inhibitors in a clinical setting without interfering with cancer treatment efficacy.

  1. Statistical analysis is missing from many figures like the majority of Fig. 1, 3, 4b, and 6. IC50 values should be reported in all dose-response curves (Fig. 2, 6, and 7).

We thank the reviewer for this remark and went over all the figures to implement statistical analysis and EC50 and IC50.

  1. Sentence in line 498 is cut off before completion.

We thank the Reviewer for bringing this to our attention and completed the paragraph. “Also, compound motor action potentials (CMAP) significantly decreased with paclitaxel treatment, but not after combinatory treatment of glycyrrhizic acid and paclitaxel, compared to control-treated animals (Figure 8c). Bodyweight (Figure 8d) and axon count (Figure 8e) did not change during treatment in either treatment group compared to control animals.”

Reviewer 2 Report

 The research  article with title Glycyrrhizic acid prevents paclitaxel-induced neuropathy via 2 inhibition of OATP-mediated neuronal uptake is well designed.

The characterized in detail the transport of paclitaxel into neurons and 519 provide evidence that inhibition of drug transporters and efficiently attenuates the neurotoxicity of paclitaxel without interfering with its antineo- plastic properties

Authors can correct the manuscript for English

Author Response

 We thank the reviewer for reading our manuscript and pointing out the errors.

1. The characterized in detail the transport of paclitaxel into neurons and 519 provide evidence that inhibition of drug transporters and efficiently attenuates the neurotoxicity of paclitaxel without interfering with its antineo- plastic properties

Sentence was updated and corrected.

2. Authors can correct the manuscript for English

We thank the reviewer for this remark. We have read the manuscript again and corrected the language and altered it for better readability.

Reviewer 3 Report

The authors in this manuscript describe a potential route of uptake of paclitaxel that leads to peripheral neuropathy. They found uptake driven by transporters, namely Organic Anion Transporting Polypeptide (OATP). The conclusion was derived from utilizing inhibitors, RT-qPCR to test expression levels and CRISPR/Cas9 knock-out cells suggested uptake via OATP1A1 and OATP1B2. The in vitro data was further supported with in vivo data, were glycyrrhizic acid prevented paclitaxel-induced neurotoxicity. Overall, the manuscript is well presented but the authors needs to address few comments:

The introduction requires revision, the current version is not well constructed for readers to quickly understand the background, hypothesis and summary of results. Line 47-48: “mitochondrial changes”, please elaborate and these lines needs more work. Line 50-55: Seems the sentence are bolded, not sure if it was an error. Line 284-285: Please add reference. Figure 1C: Consider including a line break in the x-axis to expand on the initial uptake time. Figure 3: Please include in methods section how the y-axis was calculated. Not clear from the methods section on how this was normalized.

Major Points:

1. Based on the data presented in the manuscript, have the authors considered determining the Km value of paclitaxel for OAT1A1/B2?

Author Response

  1. The introduction requires revision, the current version is not well constructed for readers to quickly understand the background, hypothesis and summary of results. 

Introduction was altered for better readability, plus hypothesis and summary was added.

„Peripheral neuropathy is one of the most common side effects of the microtubule-stabilizing chemotherapeutic agent paclitaxel, affecting up to 87% of all treated patients [1], [2]. Currently, paclitaxel, and its derivatives such as docetaxel and cephalomannine, are most commonly used to treat ovarian, breast, lung, prostate, head and neck, and gastric cancer entities [3]. In cancer cells paclitaxel induces cell death via microtubule stabilization [5], [6]; however, due to its microtubule stabilizing actions it is able to induce neuronal damage. Patients suffer predominantly from sensory symptoms, such as pain and numbness in hands and feet due to paclitaxel accumulation in the dorsal root ganglia (DRG) [13]. As the DRGs possess a more permeable blood-nerve barrier, sensory neurons in the DRG are highly susceptible to paclitaxel accumulation [14]. The paclitaxel-induced length-dependent axonal sensory neuropathy correlates with the dose, infusion time, underlying conditions, and co-treatment with other drugs [15], [16]. The influence of paclitaxel on A-β fibers gives rise to the main chemotherapy-induced peripheral neuropathy (CIPN) characteristics, such as numbness and loss of vibration sense. Dysesthesia and cold and mechanical hypersensitivity are due to the influence of paclitaxel on Aδ- and C-fibers [17].

Currently, the pathological cascade of how paclitaxel damages nerve fibers is incompletely understood. It is assumed that multimodal effects of paclitaxel cause neuronal damage, such as axonal transport impairment, mitotoxicity, and inflammation [7], [8]. In the peripheral nervous system (PNS) the microtubule stabilization affects protein, organelle, nutrient, neurotransmitter, and mRNA transport [9]. A following ATP undersupply in the periphery results in mitochondrial changes, such as swelling and vacuolization, and increased pain sensation [10]. Several inflammatory markers, such as interleukins and C-X-C motif chemokines, are thought to elicit pain sensations in patients [11]. Moreover, with the attraction of macrophages, neuronal degeneration is fostered [12].

Due to this plethora of toxic effects, there are currently no clinical treatments to prevent or treat paclitaxel-induced neuropathy. Furthermore, there is experimental evidence that different paclitaxel delivery mechanisms impact the kinetics of paclitaxel in nervous tissue and hence the degree of neurotoxicity [18], [19]. With this study we want to intervene the paclitaxel toxicity as early as possible, in this case on paclitaxel uptake into the neuronal cell. We therefore reasoned that further characterization of the neuron-specific uptake of paclitaxel may provide novel strategies to prevent peripheral neuropathy.

It is assumed that paclitaxel is actively transported into cells by organic anion-transporting polypeptides (OATPs) [20]–[22]. These encompass a family of plasma membrane transporters essential for the absorption, clearance, and tissue distribution of endogenous compounds and xenobiotics. Corresponding genes are named with the prefix SLCO, while encoding proteins are marked with OATP. Whereas the expression pattern of OATP in cancer cells and hepatocytes is well known, in sensory neurons, the overall expression profile of OATP is less well studied. An exception to this is OATP1B2 (in rodents, corresponding to OATP1B1 and OATP1B3 in humans), whose pharmacological or genetic modulation ameliorated paclitaxel-induced neuropathy in a preclinical model [23]. However, symptoms were only partially alleviated, which indicates that other transporters are likely to be involved. Moreover, OATP1B3, the human homolog of OAT1B2, has been found in ovarian and associated cancer cell lines, such as ovarian cancer cells [24], colorectal and pancreatic cancer cells [25], and castration-resistant prostate cancer cells [26]. All of which are cancer entities that are treated with paclitaxel.

This leads us to our hypothesis that organic anion-transporting polypeptides (OATPs), which are essential for the absorption and tissue distribution of endogenous and exogenous compounds, are involved in paclitaxel uptake into sensory neurons. This study identified OATP1A1 and OATP1B2 as paclitaxel transporters in sensory neurons. Further experiments in vivo showed that inhibiting paclitaxel uptake with glycyrrhizic acid could alleviate paclitaxel-induced neuropathy. The study sheds light on the mechanism of paclitaxel-induced neuropathy and identifies potential targets for prevention and treatment.”

  1. Line 47-48: “mitochondrial changes”, please elaborate and these lines needs more work. 

We thank the reviewer for this suggestion and added the mitochondria changes and overall reworked the introduction part: “A following ATP undersupply in the periphery results in mitochondrial changes, such as swelling and vacuolization, and increased pain sensation [10].”

  1. Line 50-55: Seems the sentence are bolded, not sure if it was an error.

Sentences should not be bolded, but were tried to be corrected.

  1. Line 284-285: Please add reference. 

Reference was added accordingly.

  1. Figure 1C: Consider including a line break in the x-axis to expand on the initial uptake time. 

We agree with the Reviewer and added a line brake to better visualize the initial uptake.

  1. Figure 3: Please include in methods section how the y-axis was calculated. Not clear from the methods section on how this was normalized.

Data was not normalized and is depicted as ng/µg, as in Paclitaxel in ng per µg cells. This was added as clarification in the figure description. Furthermore, explanation for the x-axis was added. “All results of LC-MS/MS were depicted as ng paclitaxel per µg cells (y-axis) correlated with the different concentrations that were applied to the cells (x-axis).”

MAJOR POINT:

Based on the data presented in the manuscript, have the authors considered determining the Km value of paclitaxel for OAT1A1/B2?

We thank the reviewer for pointing this out to us and implemented this idea accordingly. Instead of Km values the EC50 and IC50 for all graphs was added, as it seems to be a better fit as well as represent the intended value.

Round 2

Reviewer 1 Report

Concerns addressed